# Marital status differences in suicidality among transgender people

**Hui Liu [ID]1\***, **Lindsey Wilkinson2**

**1** Department of Sociology, Michigan State University, East Lansing, Michigan, United States of America,
**2** Department of Sociology, Portland State University, Portland, Oregon, United States of America

\* liuhu@msu.edu

**Data Availability Statement:** The data for this study were drawn from the 2015 U.S. Transgender Survey, which is publicly available from the following website: https://www.icpsr.umich.edu/web/ICPSR/studies/37229/datadocumentation.

## Abstract

The suicide rate for transgender people is among the highest of any group in the United States. Yet, we know little about disadvantages or resources available to transgender people to prevent suicide. The overall purpose of this study is to assess how marital status modifies the risk of suicide among transgender people. We analyzed data from the 2015 U.S. Transgender Survey to predict marital status differences in both suicide ideation and suicide attempt in the past year. The analytic sample for suicide ideation included 17,117 transgender respondents (9,182 transwomen and 7,935 transmen), and the analytic sample for suicide attempt was limited to 8,058 transgender respondents (4,342 transwomen and 3,716 transmen) who reported suicide ideation in the last 12 months. Results from binary logistic regression models suggested that never married and previously married transmen and transwomen, regardless of their partnership status, generally had higher risk of both suicide ideation and attempt than their married transgender counterparts with only one exception: never married transwomen had lower risk of suicide ideation (but not attempt) than their married transwomen counterpart after sociodemographic characteristics were accounted for. These findings draw attention to the heterogeneity of the transgender population, highlighting marital status as a key social factor in stratifying the life experiences of transgender people.

## Introduction

Suicide is the 2nd leading cause of death among 10-34-year-olds and 10th among the overall U.S. population, claiming the lives of over 47,000 Americans annually [1]. The prevalence of suicide ideation and attempt is significantly higher among transgender people— defined broadly as individuals who deviate from the gender binary or are gender variant [2, 3]—than both heterosexual people and lesbian, gay, and bisexual people [4]. According to a recent national report by the Williams Institute [4], 81.7 percent of transgender Americans reported ever seriously thinking about suicide and 40.4 percent reported ever attempting suicide at some point in their lifetimes. These numbers are in staggering contrast to 4.8% for suicide ideation and 0.6% for suicide attempt among the general U.S. population [1]. Although researchers and policymakers have called for increased attention to the unique challenges and

**Funding:** This research was partially supported by the National Institute on Aging, Grant R01AG061118. The funders had no role in study design, data collection and analysis, decision to publish, or preparation of the manuscript.

**Competing interests:** The authors have declared that no competing interests exist.

disadvantages that transgender people experience [2, 5, 6], we know little about such disadvantages or resources available to transgender people to prevent suicide [5, 7, 8].

Research among the cisgender population (i.e., those whose gender identity and sex at birth are in agreement) has long documented that married people are less likely to commit suicide than unmarried people [9–11]. Scholars assume that marriage among transgender individuals is also likely associated with individual well-being [12, 13], and advocates contend that marriage equality may reduce the disadvantages of gender and sexual minorities [14]. Yet such claims have been made with limited empirical support, especially in regard to suicide. We analyze one of the few large-scale datasets incorporating transgender people, the United States Transgender Survey (USTS), to assess whether the risk of suicide among transgender people varies by marital status. Our results speak to the implications of marriage equality for the well-being of gender minorities and to our general understanding of transgender people, one of the least understood segments of the gender- and sexual-minority population.

## Marital status and suicide

In his classic study of *Suicide* [9], Durkheim found that suicide occurred more often among some social groups than others, with unmarried people showing higher suicide risk than married people. Durkheim theorized that the underlying reason for suicide relates, for the most part, to different levels of social integration across social groups. He concluded that the more socially integrated and connected an individual is, the less likely he or she is to commit suicide. Marriage increases access to social support (i.e., the commitment, caring, advice and aid provided in personal relationship) and social integration (i.e., feeling connected with others), and married people experience higher levels of social support and social integration relative to unmarried people [15, 16]—and this is what has been theorized to explain the lower suicide rate of married people relative to unmarried people [9]. However, this perspective has been developed and tested primarily among cisgender different-gender marriages, and it is unclear whether it also holds for transgender people.

## A minority stress perspective on marital advantage in suicidality among transgender people

The minority stress paradigm was developed to address the stresses that accrue to gender and sexual minorities as a result of higher rates of stigma and discrimination [17]. According to the minority stress theory, gender- and sexual-minority status is a fundamental cause of health disparities because it is socially stigmatized and entails disadvantage [5, 17]. Transgender people represent one of the least understood and most stigmatized groups in the lesbian, gay, bisexual, and transgender (LGBT) population [17]. This "minority among minorities" experience not only boosts transgender people's stress and increases the risk of suicide by exposing them to institutional and interpersonal stigma and discrimination but also may modify how marriage shapes their experiences of social support and social integration as we discuss next.

The marital relationship has historically been recognized as the most important social relationship in adulthood and an essential pathway to a meaningful life and to personal maturity [18]. Married people have an advantage over unmarried people in access to social support and social integration [15, 16, 19]. Married transgender people may use these advantages to avoid and respond to transgender-based discrimination—a risk factor for suicide. Because transgender people often feel socially isolated and report a lack of support [20], even from their biological family and intimate partners [21], they may feel a lack of security and safety as a result of high levels of transphobia, thus increasing risk of suicide. Having a spouse as a source of social support and companionship may mitigate perceptions of isolation and reduce the risk of

suicide. Alternatively, transgender people who are more socially integrated and have greater access to social support, and thus have lower risk of suicide, may be advantaged in the marriage market and thus may be more likely to marry. Taken together, *we hypothesize that married transgender people will have lower risk of suicide ideation and attempt than unmarried transgender people.*

### Gender differences

Health and well-being benefits of marriage are suggested to be greater for men than for women because women in heterosexual marriages more often provide emotional and social support and maintain network connections in the marriage while men are more likely to receive such health promotion benefits from their wives [9, 15, 22, 23]. Yet, it is unclear whether this gendered pattern holds for transgender spouses. This pattern may be reproduced as transgender individuals seek gender coherence within relationships with different-gender partners through traditional enactments of masculinity and femininity [24, 25]. However, it is also likely that this gendered pattern diverges in marriages involving a transgender spouse because of transgender individuals' complex experiences of gender [26, 27]: research suggests that relationships involving sexual and gender minority individuals often feature less intensely gendered differences in their relationship [26, 28], particularly in same-gender marriages [29].

We expect that transwomen experience greater benefits from marriage associated with suicidality than transmen for several reasons. First, transitioning for transmen is often a "status-boosting" process resulting in the accrual of male privileges in society (e.g., labor market) [24, 27]. In contrast, transwomen may experience the double burden of being both transgender and female [27], resulting in elevated stress and increased risk of suicide ideation and attempt. Moreover, transitioning evokes more experiences of discrimination [30], violence (especially hate crimes) [31, 32], and minority stress for transwomen than transmen; transwomen may therefore feel more socially isolated than transmen. Married transwomen with a supportive spouse may feel more connected with others, less socially isolated, and more self-confident in their chosen gender [33], and thus may have lower risk of suicide ideation and attempt than unmarried transwomen. Taken together, *we expect that marital status differences in suicide ideation and attempt will differ between transwomen and transmen.*

## Materials and methods

### Data and sample

We used data from the 2015 U.S. Transgender Survey (USTS), conducted by the National Center for Transgender Equality and the National Gay and Lesbian Task Force. The USTS is the largest anonymous online survey on the experiences of transgender adults (ages 18 and older) in the United States, with 27,715 respondents from all fifty states, the District of Columbia, American Samoa, Guam, Puerto Rico [34]. Yet, USTS is not population-based or representative, as the sample was recruited using convenience-sampling techniques including venue-based sampling and snowball sampling. The survey was announced through a network of more than 800 transgender-led or transgender-serving community-based organizations and 150 active online community listservs in the United States. The total USTS sample included 27,715 valid respondents, including 17,188 (62%) transgender, 9,769 (35%) gender nonconforming, and 758 (3%) cross-dresser respondents. Because our research questions focused on transgender people, our analytic sample was restricted to those 17,188 respondents who self-identified as transgender (i.e., assigned birth sex is different from current primary gender identity and/or respondent identifies primarily as transgender or transsexual). We further excluded the 50 transgender respondents missing on marital status and the 21 respondents missing on

the suicide ideation or attempt, leaving a sample of 17,117 respondents (9,182 transwomen, i.e., assigned birth sex is male and the current primary gender identity is woman and/or respondent identifies primarily as transwoman and 7,935 transmen, i.e., assigned birth sex is female and the current primary gender identity is man and/or respondent identifies primarily as transman). The analytic sample used to estimate suicide attempt is limited to respondents who reported suicide ideation in the last 12 months, leaving 8,058 respondents (4,342 transwomen and 3,716 transmen). We used the Multiple Imputation using Chained Equations (MICE) system [35], including the dependent variables for imputation as recommended [36] to impute all other covariates.

## Measures

**Suicidality.**   We analyzed two suicidality variables reflecting suicide ideation and suicide attempt in the past year. *Suicide ideation* was measured based on the question: "At any time in the past 12 months did you seriously think about trying to kill yourself?" (1 = yes, 0 = no). Respondents who answered "yes" to this question were then asked about *suicide attempt*: "During the past 12 months, did you try to kill yourself?" (1 = yes, 0 = no).

**Marital status.**   Marital status was measured based on two questions: "What is your current legal marital status?" and "What is your current relationship status?". Because prior research suggests that never married and previously married exhibit very different health patterns when compared to the married, with further distinctions by partnership status [13, 15, 19], we created the following five marital status groups: 1) married, including respondents with a legally recognized civil union or registered domestic partnership (the reference group), 2) never married and currently unpartnered; 3) never married and currently partnered (living together or not), 4) previously married (including separated, divorced, and widowed) and currently unpartnered; and 5) previously married and currently partnered.

**Covariates.**   We included two types of control covariates: sociodemographic covariates and transition-stage indicators. Sociodemographic covariates included: *Age* at the time of the survey was measured in years, ranging from 18 to 85. *Educational attainment* was measured as the highest degree attained, including less than a high school diploma, a high school diploma, some college, a college degree (the reference group), and an advanced degree. *Race and ethnicity* included non-Latinx White (reference), non-Latinx Black or African American, non-Latinx Asian or Pacific Islander, non-Latinx American Indian or Alaska Native, Hispanic or Latinx and multiracial. *Sexual orientation* included straight (reference), lesbian/gay, bisexual, and asexual and other. *Geographic region* included Northeast, Midwest, South, and West (reference). *Employment status* was measured with a dichotomous variable indicating whether the respondent was currently employed (1 = yes, 0 = no). *Household income* was measured as total combined household income in 2014 (except food stamps or WIC), ranging from 1 to 18, where 1 = no income and 18 = $150,000 or more. We also controlled for whether or not the respondents owned their home (1 = yes, 0 = no) and for *the number of minor children* in the respondent's household at the time of the survey (range 0–9). *Health insurance coverage* indicated whether respondent was covered by any health insurance or health coverage plan at the time of the survey (1 = yes, 0 = no).

The stage of transition from one's assigned sex or gender to one's chosen sex or gender has important implications for the experiences of transgender people [31, 37–40]. Because both marital status and suicide risk likely vary by transition stage [31, 37–40], we controlled for the following indicators of transition stage: the *age respondent first started telling others they were transgender*, even if they did not use the term transgender (range from 1–98); respondent's *full-time transgender status*, indicated by whether the respondent was living full-time in a

gender different from the one assigned at birth at the time of the survey (1 = yes, 0 = no); *visual conformity*, based on a question asking how often respondent believes people can tell they are transgender, even if the respondent does not disclose their transgender status (range from 1 = always to 5 = never); and *medical transition*, indicated by whether the respondent had begun a medical transition through hormones or surgery (1 = yes, 0 = no). We also controlled for how many people in the respondents' lives knew they were transgender, using the USTS constructed measure *"outness"* (range from 1 = none to 4 = all).

## Statistical methods

We estimated binary logistic regression models to analyze suicide ideation and suicide attempt separately. We ran three models for each suicidality outcome. Model 1 tested the basic association between marital status and suicidality without controlling for any covariates. In Model 2, we added controls for basic sociodemographic covariates to assess whether sociodemographic characteristics contributed to these marital status differences in suicidality among transgender people. In Model 3, transition stage covariates were added to assess whether transition stage further explained any associations. To fully consider potential gender differences, we stratified all analyses by transwomen and transmen. We ran all analyses using Stata 15 [41]. All analyses are weighted with the standard survey weight ("surveyweight"), making the USTS sample more representative of the U.S. population in terms of race/ethnicity and age [34]. The standard survey weight incorporates a weight constructed using information from the American Community Survey to help adjust for hypothesized over-representation of whites and the 18-year old group in the USTS sample.

## Results

### Descriptive results

Table 1 shows the weighted descriptive statistics of all analyzed variables by marital status for transwomen and transmen. For both transmen and transwomen, all unmarried groups including both the never married and previously married regardless of partnership status had higher proportions of suicide ideation and attempt in the past year than the married (although some differences were not statistically significant, see details in Table 1). In general, married transmen and transwomen were older than their never married counterparts and younger than their previously married counterparts; married transmen and transwomen were more likely to be white, employed and homeowner with more minor children, and they were more likely to have college or advantaged degree, health insurance coverage, higher household income and higher levels of visual conformity than their never married and previously married counterparts.

### Logistic regression results

Table 2 shows logistic regression results for suicide ideation and suicide attempt for transmen. The results from Model 1 of Table 2 suggest that without controlling for any covariates, never married transmen, either unpartnered (OR = 2.82, p = .001) or partnered (OR = 2.43, p = .001), and previously married transmen who were unpartnered (OR = 1.66, p = .004) had significantly higher odds of suicide ideation than married transmen. After basic demographic covariates were held consistent in Model 2, the difference in suicide ideation between never married, partnered transmen and married transmen was reduced to insignificance (OR = 1.20, p = .095); the difference between unpartnered never married transmen and married transmen was also reduced in size but remained significant (OR = 1.38, p = .004); while the difference

**Table 1. Weighted descriptive statistics by marital status and gender, transmen.**

| | Married | Never married, partnered | | Never married, unpartnered | | Sep/div/ widowed, partnered | | Sep/div/widowed, unpartnered | |
|---|---|---|---|---|---|---|---|---|---|
| | | Transmen | | | | | | | |
| Suicidal ideation | 0.27 | 0.47 | *** | 0.51 | *** | 0.32 | | 0.38 | ** |
| Suicide attempt | 0.11 | 0.15 | | 0.11 | | 0.19 | | 0.12 | |
| *Sociodemographic Covariates* | | | | | | | | | |
| Age | 36.14 | 26.12 | *** | 26.38 | *** | 37.01 | | 41.48 | *** |
| Educational attainment | | | | | | | | | |
| Less than HS diploma | 0.02 | 0.02 | | 0.02 | | 0.01 | | 0.03 | |
| HS diploma | 0.05 | 0.12 | *** | 0.13 | *** | 0.06 | | 0.09 | |
| Some college | 0.24 | 0.44 | *** | 0.43 | *** | 0.37 | ** | 0.28 | |
| Associate's degree | 0.10 | 0.08 | | 0.08 | | 0.15 | | 0.12 | |
| Bachelor's degree | 0.26 | 0.21 | ** | 0.24 | | 0.26 | | 0.23 | |
| Advanced degree | 0.33 | 0.11 | *** | 0.10 | *** | 0.15 | *** | 0.25 | * |
| Race/ethnicity | | | | | | | | | |
| White | 0.65 | 0.58 | ** | 0.57 | *** | 0.58 | | 0.61 | |
| Black | 0.13 | 0.13 | | 0.16 | | 0.19 | | 0.18 | |
| Latinx | 0.16 | 0.22 | * | 0.18 | | 0.19 | | 0.15 | |
| Asian/Pacific Islander | 0.01 | 0.01 | * | 0.01 | *** | 0.01 | | 0.01 | |
| American Indian/Alaskan Native | 0.02 | 0.04 | | 0.06 | | 0.01 | | 0.02 | |
| Multiracial | 0.03 | 0.03 | | 0.02 | | 0.02 | | 0.02 | |
| Sexual orientation | | | | | | | | | |
| Straight | 0.33 | 0.22 | *** | 0.20 | *** | 0.28 | | 0.31 | |
| Gay/lesbian | 0.12 | 0.10 | | 0.13 | | 0.14 | | 0.21 | ** |
| Bisexual | 0.39 | 0.39 | | 0.35 | * | 0.30 | | 0.24 | *** |
| Asexual/other | 0.14 | 0.25 | *** | 0.27 | *** | 0.23 | ** | 0.18 | |
| Region | | | | | | | | | |
| Northeast | 0.24 | 0.21 | | 0.22 | | 0.13 | *** | 0.17 | * |
| Midwest | 0.14 | 0.20 | *** | 0.19 | ** | 0.13 | | 0.14 | |
| South | 0.27 | 0.28 | | 0.31 | | 0.36 | * | 0.32 | |
| West | 0.35 | 0.31 | | 0.29 | ** | 0.36 | | 0.38 | |
| Employed | 0.80 | 0.72 | *** | 0.66 | *** | 0.75 | | 0.73 | |
| Household income | 13.17 | 9.31 | *** | 8.99 | *** | 11.46 | *** | 10.47 | *** |
| Homeowner | 0.38 | 0.09 | *** | 0.06 | *** | 0.20 | *** | 0.19 | *** |
| Number of minor children | 0.49 | 0.27 | *** | 0.24 | *** | 0.51 | | 0.49 | |
| Health insurance | 0.91 | 0.86 | ** | 0.87 | ** | 0.87 | | 0.83 | ** |
| *Transition Stage Covariates* | | | | | | | | | |
| Age told others transgender | 22.64 | 19.04 | *** | 19.18 | *** | 23.36 | | 26.02 | ** |
| Live full time as different gender | 0.89 | 0.83 | *** | 0.78 | *** | 0.89 | | 0.87 | |
| Visual conformity | 2.66 | 2.45 | *** | 2.51 | *** | 2.49 | ** | 2.57 | |
| Medical transition | 0.88 | 0.68 | *** | 0.66 | *** | 0.83 | | 0.87 | |
| Outness | 2.84 | 2.79 | * | 2.75 | *** | 2.84 | | 2.96 | ** |
| N | 1185 | 2842 | | 3400 | | 205 | | 303 | |
| Suicidal ideation | 0.38 | 0.52 | *** | 0.51 | *** | 0.46 | ** | 0.40 | |
| Suicide attempt | 0.11 | 0.19 | ** | 0.19 | *** | 0.14 | | 0.20 | ** |
| *Sociodemographic Covariates* | | | | | | | | | |
| Age | 46.68 | 28.65 | *** | 30.59 | *** | 46.81 | | 51.23 | *** |

(*Continued*)

**Table 1.** (Continued)

| | Transmen | | | | | | | | | |
|---|---|---|---|---|---|---|---|---|---|---|
| | Married | Never married, partnered | | Never married, unpartnered | | Sep/div/widowed, partnered | | Sep/div/widowed, unpartnered | |
| Educational attainment | | | | | | | | | |
| Less than HS diploma | 0.02 | 0.04 | * | 0.04 | | 0.01 | | 0.02 | |
| HS diploma | 0.07 | 0.12 | *** | 0.15 | *** | 0.09 | | 0.10 | * |
| Some college | 0.25 | 0.42 | *** | 0.41 | *** | 0.33 | ** | 0.27 | |
| Associate's degree | 0.12 | 0.08 | ** | 0.10 | | 0.17 | * | 0.16 | * |
| Bachelor's degree | 0.30 | 0.25 | * | 0.22 | *** | 0.22 | ** | 0.26 | |
| Advanced degree | 0.23 | 0.08 | *** | 0.08 | *** | 0.17 | ** | 0.19 | * |
| Race/ethnicity | | | | | | | | | |
| White | 0.79 | 0.63 | *** | 0.56 | *** | 0.78 | | 0.79 | |
| Black | 0.07 | 0.13 | ** | 0.16 | *** | 0.05 | | 0.05 | |
| Latinx | 0.09 | 0.15 | ** | 0.20 | *** | 0.11 | | 0.11 | |
| Asian/Pacific Islander | 0.01 | 0.01 | *** | 0.00 | * | 0.02 | | 0.01 | |
| American Indian/Alaskan Native | 0.03 | 0.06 | | 0.05 | * | 0.02 | * | 0.02 | |
| Multiracial | 0.02 | 0.02 | | 0.02 | | 0.02 | | 0.01 | |
| Sexual orientation | | | | | | | | | |
| Straight | 0.16 | 0.17 | | 0.21 | ** | 0.11 | * | 0.18 | |
| Gay/lesbian | 0.34 | 0.25 | *** | 0.24 | *** | 0.36 | | 0.27 | *** |
| Bisexual | 0.27 | 0.26 | | 0.26 | | 0.28 | | 0.27 | |
| Asexual/other | 0.18 | 0.26 | *** | 0.23 | ** | 0.20 | | 0.22 | ** |
| Region | | | | | | | | | |
| Northeast | 0.16 | 0.19 | | 0.20 | ** | 0.13 | | 0.15 | |
| Midwest | 0.21 | 0.20 | | 0.17 | ** | 0.20 | | 0.20 | |
| South | 0.30 | 0.32 | | 0.31 | | 0.29 | | 0.29 | |
| West | 0.32 | 0.28 | * | 0.31 | | 0.37 | | 0.35 | |
| Employed | 0.75 | 0.67 | *** | 0.61 | *** | 0.71 | | 0.64 | *** |
| Household income | 14.07 | 9.48 | *** | 8.97 | *** | 12.03 | *** | 11.32 | *** |
| Homeowner | 0.60 | 0.08 | *** | 0.08 | *** | 0.32 | *** | 0.32 | *** |
| Number of minor children | 0.55 | 0.20 | *** | 0.19 | *** | 0.28 | *** | 0.23 | *** |
| Health insurance | 0.93 | 0.83 | *** | 0.81 | *** | 0.86 | *** | 0.87 | *** |
| *Transition Stage Covariates* | | | | | | | | | |
| Age told others transgender | 31.17 | 20.40 | *** | 21.48 | *** | 28.71 | ** | 33.11 | *** |
| Live full time as different gender | 0.58 | 0.74 | *** | 0.63 | ** | 0.79 | *** | 0.80 | *** |
| Visual conformity | 2.31 | 2.31 | | 2.29 | | 2.26 | | 2.19 | *** |
| Medical transition | 0.79 | 0.77 | | 0.71 | *** | 0.90 | *** | 0.89 | *** |
| Outness | 2.76 | 2.89 | *** | 2.78 | | 3.01 | *** | 3.03 | *** |
| N | 1968 | 1835 | | 3257 | | 598 | | 1524 | |

*** p<0.001

** p<0.01

* p<0.05: Two-tailed t-tests for continuous variables and tests of proportions for categorical variables, compared to the married group.

between previously married, unpartnered transmen and married transmen remained, with little change (OR = 1.68, p = .008). In Model 3, after the transition stage indicators were added, the differences in suicide ideation for never married, unpartnered transmen (OR = 1.32,

**Table 2. Estimated odds ratios and 95% CIs from logistic regression models predicting suicidality, transmen.**

| | Suicide Ideation | | | | | | | | |
| --- | --- | --- | --- | --- | --- | --- | --- | --- | --- |
| | Model 1 | | | Model 2 | | | Model 3 | | |
| | OR | 95% CI | p | OR | 95% CI | p | OR | 95% CI | p |
| *Marital status (0 = currently married)* | | | | | | | | | |
| Never married, unpartnered | 2.82 | 2.33–3.41 | 0.000 | 1.38 | 1.11–1.71 | 0.004 | 1.32 | 1.06–1.65 | 0.013 |
| Never married, partnered | 2.43 | 1.99–2.96 | 0.000 | 1.20 | 0.97–1.49 | 0.095 | 1.14 | 0.92–1.42 | 0.232 |
| Previously married, unpartnered | 1.66 | 1.17–2.34 | 0.004 | 1.68 | 1.14–2.48 | 0.008 | 1.66 | 1.13–2.43 | 0.010 |
| Previously married, partnered | 1.26 | 0.84–1.88 | 0.258 | 1.06 | 0.69–1.63 | 0.782 | 1.02 | 0.67–1.55 | 0.927 |
| *Sociodemographic covariates* | | | | | | | | | |
| Age | | | | 0.96 | 0.95–0.97 | 0.000 | 0.97 | 0.96–0.98 | 0.000 |
| Race/ethnicity (0 = Non-Latinx white) | | | | | | | | | |
| Non-Latinx Black | | | | 0.99 | 0.75–1.31 | 0.944 | 1.08 | 0.82–1.42 | 0.590 |
| Latinx | | | | 0.85 | 0.69–1.05 | 0.125 | 0.88 | 0.71–1.08 | 0.223 |
| Native American | | | | 1.79 | 1.19–2.69 | 0.005 | 1.68 | 1.12–2.51 | 0.012 |
| Asian | | | | 0.83 | 0.60–1.13 | 0.229 | 0.83 | 0.60–1.15 | 0.267 |
| Multiracial | | | | 1.10 | 0.89–1.36 | 0.388 | 1.12 | 0.90–1.40 | 0.306 |
| Educational attainment (0 = 4-year degree) | | | | | | | | | |
| Less than high school | | | | 2.27 | 1.33–3.84 | 0.002 | 2.11 | 1.23–3.63 | 0.007 |
| High school | | | | 1.90 | 1.49–2.42 | 0.000 | 1.70 | 1.32–2.18 | 0.000 |
| Some college | | | | 1.47 | 1.23–1.74 | 0.000 | 1.39 | 1.17–1.66 | 0.000 |
| Associate's degree | | | | 1.16 | 0.88–1.52 | 0.288 | 1.15 | 0.87–1.51 | 0.327 |
| Advanced degree | | | | 0.91 | 0.73–1.15 | 0.444 | 0.93 | 0.74–1.17 | 0.515 |
| Sexual orientation (0 = Straight) | | | | | | | | | |
| Gay/lesbian | | | | 1.18 | 0.94–1.47 | 0.144 | 1.07 | 0.85–1.33 | 0.570 |
| Bisexual/queer | | | | 1.25 | 1.06–1.49 | 0.010 | 1.22 | 1.02–1.45 | 0.027 |
| Asexual/other | | | | 1.46 | 1.20–1.77 | 0.000 | 1.31 | 1.07–1.59 | 0.008 |
| Employed | | | | 0.79 | 0.68–0.92 | 0.003 | 0.84 | 0.72–0.98 | 0.030 |
| Household income | | | | 0.98 | 0.96–0.99 | 0.001 | 0.98 | 0.96–0.99 | 0.005 |
| Homeowner | | | | 0.76 | 0.61–0.95 | 0.017 | 0.76 | 0.61–0.95 | 0.014 |
| Health insurance | | | | 0.72 | 0.58–0.88 | 0.001 | 0.75 | 0.61–0.93 | 0.008 |
| Number of minor children | | | | 1.05 | 0.95–1.16 | 0.325 | 1.03 | 0.94–1.14 | 0.503 |
| Region (0 = West) | | | | | | | | | |
| Northeast | | | | 1.09 | 0.90–1.31 | 0.386 | 1.08 | 0.89–1.30 | 0.425 |
| Midwest | | | | 1.24 | 1.03–1.50 | 0.024 | 1.20 | 1.00–1.45 | 0.053 |
| South | | | | 1.20 | 1.00–1.44 | 0.052 | 1.14 | 0.95–1.38 | 0.152 |
| *Transition stage covariates* | | | | | | | | | |
| Age told others transgender | | | | | | | 0.99 | 0.98–1.00 | 0.053 |
| Live full time as different gender | | | | | | | 0.90 | 0.72–1.13 | 0.364 |
| Medical transition | | | | | | | 0.63 | 0.52–0.76 | 0.000 |
| Visual conformity | | | | | | | 0.76 | 0.69–0.85 | 0.000 |
| Outness | | | | | | | 1.03 | 0.89–1.18 | 0.689 |
| Constant | 0.37 | 0.31–0.44 | 0.000 | 2.38 | 1.44–3.90 | 0.001 | 6.22 | 3.07–12.60 | 0.000 |
| N | 7935 | | | | | | | | |
| | Suicide Attempt | | | | | | | | |
| | Model 1 | | | Model 2 | | | Model 3 | | |
| | OR | 95% CI | p | OR | 95% CI | p | OR | 95% CI | p |
| *Marital status (0 = currently married)* | | | | | | | | | |
| Never married, unpartnered | 1.43 | 0.86–2.38 | 0.171 | 0.97 | 0.56–1.67 | 0.900 | 0.96 | 0.57–1.62 | 0.886 |

*(Continued)*

**Table 2.** (Continued)

| | | | | | | | | | |
|---|---|---|---|---|---|---|---|---|---|
| Never married, partnered | 1.46 | 0.86–2.46 | 0.157 | 0.94 | 0.54–1.64 | 0.837 | 0.89 | 0.53–1.51 | 0.663 |
| Previously married, unpartnered | 1.10 | 0.45–2.72 | 0.835 | 1.09 | 0.49–2.44 | 0.836 | 0.91 | 0.41–2.05 | 0.827 |
| Previously married, partnered | 1.85 | 0.67–5.13 | 0.236 | 1.95 | 0.72–5.26 | 0.188 | 1.72 | 0.64–4.64 | 0.285 |
| *Sociodemographic covariates* | | | | | | | | | |
| Age | | | | 0.98 | 0.96–1.00 | 0.077 | 0.99 | 0.96–1.01 | 0.295 |
| Race/ethnicity (0 = Non-Latinx white) | | | | | | | | | |
| Non-Latinx Black | | | | 1.62 | 0.97–2.73 | 0.068 | 1.59 | 0.94–2.69 | 0.087 |
| Latinx | | | | 1.31 | 0.88–1.94 | 0.177 | 1.36 | 0.92–2.00 | 0.126 |
| Native American | | | | 2.35 | 1.26–4.38 | 0.007 | 1.96 | 1.01–3.80 | 0.047 |
| Asian | | | | 2.11 | 1.19–3.73 | 0.010 | 2.19 | 1.21–3.95 | 0.010 |
| Multiracial | | | | 1.72 | 1.16–2.54 | 0.007 | 1.71 | 1.15–2.53 | 0.008 |
| Educational attainment (0 = 4-year degree) | | | | | | | | | |
| Less than high school | | | | 2.09 | 1.06–4.13 | 0.034 | 1.88 | 0.95–3.71 | 0.068 |
| High school | | | | 2.23 | 1.35–3.69 | 0.002 | 2.05 | 1.22–3.46 | 0.007 |
| Some college | | | | 1.53 | 0.97–2.41 | 0.068 | 1.49 | 0.94–2.38 | 0.092 |
| Associate's degree | | | | 1.55 | 0.83–2.89 | 0.173 | 1.53 | 0.81–2.88 | 0.189 |
| Advanced degree | | | | 0.72 | 0.33–1.54 | 0.395 | 0.74 | 0.34–1.60 | 0.441 |
| Sexual orientation (0 = Straight) | | | | | | | | | |
| Gay/lesbian | | | | 0.77 | 0.48–1.24 | 0.277 | 0.81 | 0.51–1.30 | 0.389 |
| Bisexual/queer | | | | 0.84 | 0.58–1.23 | 0.372 | 0.91 | 0.63–1.32 | 0.628 |
| Asexual/other | | | | 1.23 | 0.86–1.76 | 0.248 | 1.28 | 0.90–1.83 | 0.173 |
| Employed | | | | 0.96 | 0.73–1.27 | 0.771 | 0.98 | 0.74–1.29 | 0.862 |
| Household income | | | | 0.97 | 0.94–1.00 | 0.052 | 0.97 | 0.95–1.00 | 0.068 |
| Homeowner | | | | 1.20 | 0.60–2.42 | 0.605 | 1.13 | 0.60–2.13 | 0.706 |
| Health insurance | | | | 0.54 | 0.38–0.77 | 0.001 | 0.56 | 0.39–0.79 | 0.001 |
| Number of minor children | | | | 1.04 | 0.88–1.23 | 0.661 | 1.06 | 0.90–1.24 | 0.496 |
| Region (0 = West) | | | | | | | | | |
| Northeast | | | | 1.14 | 0.75–1.72 | 0.536 | 1.09 | 0.72–1.65 | 0.685 |
| Midwest | | | | 1.40 | 0.96–2.03 | 0.083 | 1.39 | 0.95–2.05 | 0.091 |
| South | | | | | 0.81–1.69 | 0.399 | 1.21 | 0.84–1.74 | 0.307 |
| *Transition stage covariates* | | | | | | | | | |
| Age told others transgender | | | | | | | 0.98 | 0.95–1.00 | 0.038 |
| Live full time as different gender | | | | | | | 2.00 | 1.36–2.96 | 0.000 |
| Medical transition | | | | | | | 0.63 | 0.45–0.90 | 0.011 |
| Visual conformity | | | | | | | 0.93 | 0.76–1.14 | 0.482 |
| Outness | | | | | | | 1.29 | 0.97–1.72 | 0.082 |
| Constant | 0.12 | 0.08–0.20 | 0.000 | 0.30 | 0.10–0.92 | 0.035 | 0.16 | 0.03–0.77 | 0.022 |
| N | | | | 3716 | | | | | |

p = .013) and previously married, unpartnered transmen (OR = 1.66, p = .010), in comparison to married transmen, remained statistically significant. Results in Table 2 suggest no significant difference in suicide attempt across marital status groups among transmen in all models.

Table 3 shows logistic regression results for transwomen. The results from Model 1 of Table 3 suggest that without controlling for any covariates, never married transwomen, either unpartnered (OR = 1.75, p = .001) or partnered (OR = 1.78, p = .001), and previously married transwomen who were partnered (OR = 1.44, p = .001) had significantly higher odds of suicide ideation than married transwomen. Interestingly, after basic demographic covariates were hold consistent in Model 2, the odds ratios of suicide ideation were reversed among never

**Table 3. Estimated odds ratios and 95% CIs from logistic regression models predicting suicidality, transwomen.**

| | Suicide Ideation | | | | | | | | |
|---|---|---|---|---|---|---|---|---|---|
| | Model 1 | | | Model 2 | | | Model 3 | | |
| | OR | 95% CI | p | OR | 95% CI | p | OR | 95% CI | p |
| *Marital status (0 = currently married)* | | | | | | | | | |
| Never married, unpartnered | 1.75 | 1.51–2.02 | 0.000 | 0.72 | 0.60–0.87 | 0.001 | 0.73 | 0.60–0.87 | 0.001 |
| Never married, partnered | 1.78 | 1.52–2.09 | 0.000 | 0.70 | 0.57–0.85 | 0.000 | 0.70 | 0.57–0.86 | 0.001 |
| Previously married, unpartnered | 1.12 | 0.95–1.32 | 0.176 | 1.13 | 0.94–1.36 | 0.193 | 1.13 | 0.94–1.36 | 0.200 |
| Previously married, partnered | 1.44 | 1.15–1.79 | 0.001 | 1.29 | 1.02–1.62 | 0.037 | 1.30 | 1.02–1.64 | 0.032 |
| *Sociodemographic covariates* | | | | | | | | | |
| Age | | | | 0.96 | 0.96–0.97 | 0.000 | 0.96 | 0.95–0.97 | 0.000 |
| Race/ethnicity (0 = Non-Latinx white) | | | | | | | | | |
| Non-Latinx Black | | | | 0.58 | 0.44–0.77 | 0.000 | 0.60 | 0.45–0.79 | 0.000 |
| Latinx | | | | 1.06 | 0.85–1.31 | 0.603 | 1.07 | 0.86–1.33 | 0.540 |
| Native American | | | | 0.96 | 0.64–1.46 | 0.853 | 0.97 | 0.64–1.47 | 0.876 |
| Asian | | | | 0.64 | 0.48–0.85 | 0.002 | 0.66 | 0.49–0.88 | 0.004 |
| Multiracial | | | | 1.12 | 0.88–1.42 | 0.368 | 1.13 | 0.89–1.44 | 0.327 |
| Educational attainment (0 = 4-year degree) | | | | | | | | | |
| Less than high school | | | | 1.35 | 0.86–2.11 | 0.190 | 1.38 | 0.87–2.18 | 0.172 |
| High school | | | | 1.09 | 0.89–1.33 | 0.389 | 1.10 | 0.91–1.35 | 0.329 |
| Some college | | | | 1.12 | 0.96–1.29 | 0.144 | 1.12 | 0.97–1.30 | 0.123 |
| Associate's degree | | | | 1.00 | 0.82–1.22 | 0.981 | 1.01 | 0.83–1.22 | 0.961 |
| Advanced degree | | | | 0.92 | 0.77–1.11 | 0.401 | 0.93 | 0.77–1.11 | 0.420 |
| Sexual orientation (0 = Straight) | | | | | | | | | |
| Gay/lesbian | | | | 1.06 | 0.90–1.25 | 0.488 | 1.04 | 0.88–1.23 | 0.663 |
| Bisexual/queer | | | | 1.18 | 1.00–1.40 | 0.051 | 1.15 | 0.97–1.37 | 0.098 |
| Asexual/other | | | | 1.52 | 1.27–1.81 | 0.000 | 1.47 | 1.24–1.76 | 0.000 |
| Employed | | | | 0.78 | 0.68–0.90 | 0.000 | 0.78 | 0.68–0.89 | 0.000 |
| Household income | | | | 0.97 | 0.96–0.98 | 0.000 | 0.97 | 0.96–0.98 | 0.000 |
| Homeowner | | | | 0.82 | 0.70–0.95 | 0.008 | 0.82 | 0.71–0.95 | 0.009 |
| Health insurance | | | | 0.87 | 0.74–1.03 | 0.106 | 0.86 | 0.73–1.02 | 0.083 |
| Number of minor children | | | | 1.08 | 0.99–1.17 | 0.077 | 1.08 | 1.00–1.18 | 0.063 |
| Region (0 = West) | | | | | | | | | |
| Northeast | | | | 1.04 | 0.89–1.23 | 0.610 | 1.05 | 0.89–1.24 | 0.566 |
| Midwest | | | | 1.06 | 0.91–1.23 | 0.465 | 1.07 | 0.92–1.24 | 0.396 |
| South | | | | 0.94 | 0.81–1.09 | 0.409 | 0.95 | 0.82–1.10 | 0.519 |
| *Transition stage covariates* | | | | | | | | | |
| Age told others transgender | | | | | | | 1.00 | 1.00–1.01 | 0.702 |
| Live full time as different gender | | | | | | | 0.90 | 0.77–1.05 | 0.193 |
| Medical transition | | | | | | | 1.17 | 0.98–1.40 | 0.080 |
| Visual conformity | | | | | | | 0.86 | 0.79–0.93 | 0.000 |
| Outness | | | | | | | 0.97 | 0.87–1.09 | 0.635 |
| Constant | 0.60 | 0.54–0.67 | 0.000 | 7.32 | 4.95–10.82 | 0.000 | 10.88 | 6.43–18.41 | 0.000 |
| N | | | | 9182 | | | | | |
| | Suicide Attempt | | | | | | | | |
| | Model 1 | | | Model 2 | | | Model 3 | | |
| | OR | 95% CI | p | OR | 95% CI | p | OR | 95% CI | p |
| *Marital status (0 = currently married)* | | | | | | | | | |
| Never married, unpartnered | 1.83 | 1.32–2.54 | 0.000 | 0.75 | 0.50–1.12 | 0.156 | 0.77 | 0.51–1.14 | 0.190 |

*(Continued)*

**Table 3.** (Continued)

| | | | | | | | | | |
|---|---|---|---|---|---|---|---|---|---|
| Never married, partnered | 1.85 | 1.29–2.66 | 0.001 | 0.75 | 0.49–1.13 | 0.167 | 0.70 | 0.46–1.06 | 0.089 |
| Previously married, unpartnered | 1.89 | 1.28–2.79 | 0.001 | 1.73 | 1.14–2.63 | 0.010 | 1.64 | 1.09–2.49 | 0.018 |
| Previously married, partnered | 1.29 | 0.79–2.13 | 0.312 | 1.11 | 0.66–1.86 | 0.698 | 1.04 | 0.62–1.74 | 0.887 |
| *Sociodemographic covariates* | | | | | | | | | |
| Age | | | | 0.97 | 0.96–0.98 | 0.000 | 0.97 | 0.95–0.98 | 0.000 |
| Race/ethnicity (0 = Non-Latinx white) | | | | | | | | | |
| Non-Latinx Black | | | | 1.69 | 1.03–2.76 | 0.038 | 1.60 | 0.97–2.63 | 0.066 |
| Latinx | | | | 1.09 | 0.74–1.59 | 0.678 | 1.11 | 0.77–1.61 | 0.580 |
| Native American | | | | 0.88 | 0.39–1.97 | 0.747 | 0.84 | 0.37–1.88 | 0.662 |
| Asian | | | | 2.17 | 1.32–3.55 | 0.002 | 2.18 | 1.31–3.61 | 0.003 |
| Multiracial | | | | 1.49 | 0.98–2.25 | 0.061 | 1.41 | 0.93–2.14 | 0.106 |
| Educational attainment (0 = 4-year degree) | | | | | | | | | |
| Less than high school | | | | 2.34 | 1.23–4.45 | 0.010 | 2.37 | 1.27–4.42 | 0.007 |
| High school | | | | 1.40 | 0.97–2.01 | 0.073 | 1.38 | 0.95–2.00 | 0.093 |
| Some college | | | | 1.02 | 0.76–1.38 | 0.881 | 1.03 | 0.77–1.40 | 0.825 |
| Associate's degree | | | | 1.15 | 0.75–1.78 | 0.524 | 1.17 | 0.76–1.79 | 0.482 |
| Advanced degree | | | | 0.81 | 0.49–1.32 | 0.394 | 0.79 | 0.48–1.31 | 0.358 |
| Sexual orientation (0 = Straight) | | | | | | | | | |
| Gay/lesbian | | | | 0.97 | 0.68–1.39 | 0.872 | 1.03 | 0.72–1.47 | 0.878 |
| Bisexual/queer | | | | 1.21 | 0.86–1.69 | 0.269 | 1.29 | 0.92–1.82 | 0.143 |
| Asexual/other | | | | 1.31 | 0.93–1.83 | 0.123 | 1.39 | 0.99–1.96 | 0.058 |
| Employed | | | | 0.61 | 0.48–0.76 | 0.000 | 0.61 | 0.49–0.77 | 0.000 |
| Household income | | | | 0.97 | 0.94–0.99 | 0.002 | 0.97 | 0.95–0.99 | 0.004 |
| Homeowner | | | | 0.92 | 0.62–1.35 | 0.668 | 0.97 | 0.66–1.42 | 0.873 |
| Health insurance | | | | 1.25 | 0.92–1.69 | 0.157 | 1.25 | 0.93–1.69 | 0.143 |
| Number of minor children | | | | 0.97 | 0.82–1.14 | 0.682 | 0.98 | 0.84–1.16 | 0.843 |
| Region (0 = West) | | | | | | | | | |
| Northeast | | | | 1.07 | 0.77–1.48 | 0.698 | 1.05 | 0.76–1.45 | 0.782 |
| Midwest | | | | 1.10 | 0.83–1.46 | 0.515 | 1.11 | 0.83–1.48 | 0.483 |
| South | | | | 0.98 | 0.74–1.29 | 0.868 | 1.00 | 0.76–1.33 | 0.990 |
| *Transition stage covariates* | | | | | | | | | |
| Age told others transgender | | | | | | | 1.00 | 0.98–1.01 | 0.673 |
| Live full time as different gender | | | | | | | 1.72 | 1.25–2.34 | 0.001 |
| Medical transition | | | | | | | 0.90 | 0.65–1.24 | 0.508 |
| Visual conformity | | | | | | | 1.03 | 0.88–1.19 | 0.744 |
| Outness | | | | | | | 1.13 | 0.90–1.41 | 0.284 |
| Constant | 0.13 | 0.10–0.17 | 0.000 | 0.72 | 0.32–1.62 | 0.427 | 0.38 | 0.13–1.14 | 0.086 |
| N | | | | | 4342 | | | | |

married transwomen, both unpartnered (OR = 0.72, p = .001) and partnered (OR = 0.70, p < .000), relative to married transwomen. The odds of suicide ideation were still significantly higher among previously married, partnered transwomen than married transwomen (OR = 1.29, p = .037) even after controlling for demographic covariates. A comparison of results from Models 2 and 3 in Table 3 suggests that adding additional controls for transition stage indicators did not change the estimated marital status differences in suicide ideation for transwomen. Specifically, the results in Model 3 of Table 3 suggest that in comparison to their married transwomen counterparts, the odds of suicide ideation in the past year were 27% (OR = 0.73, p = .001) lower for never married, unpartnered transwomen, 30% (OR = 0.70,

p = .001) lower for never married, partnered transwomen, but 30% (OR = 1.30, p = .032) higher for previously married, partnered transwomen, after all covariates (sociodemographic and transitions stage) were included.

In terms of suicide attempt, the results from Model 1 of Table 3 suggest that without controlling for any covariates, never married transwomen, either unpartnered (OR = 1.83, p = .000) or partnered (OR = 1.85, p = .001), and previously married transwomen who were unpartnered (OR = 1.89, p = .001) had significantly higher odds of suicide attempt than married transwomen. After sociodemographic covariates were controlled in Model 2, the difference in suicide attempt between never married transwomen, both unpartnered (OR = 0.75, p = .156) and partnered (OR = 0.75, p = .167), and married transwomen became insignificant. Yet the higher odds of suicide attempt among previously married, unpartnered transwomen, relative to married transwomen, remained significant (OR = 1.73, p = .010). A comparison of results from Models 2 and 3 in Table 3 suggests that adding additional controls for transition stage indicators did not change the estimated marital status differences in suicide attempt for transwomen. Specifically, as shown in Model 3 of Table 3, the odds of suicide attempt in the past year were 64.7% (OR = 1.64, p = .018) higher for previously married, unpartnered transwomen relative to their married transwomen counterparts after all covariates (sociodemographic and transitions stage) are included.

## Discussion

Despite the growing size and visibility [42, 43], transgender population remains to be understudied in scientific literature—perhaps as a result of data limitations. In this study, we analyze one of the first and most comprehensive large-scale samples of transgender people in the United States to assess how marital status modifies the risk of suicide among transgender people. Our results suggest significant heterogeneity in suicide risk across marital status groups among transmen and transwomen. The sizes of marital status effects are comparable to other fundamental factors of social determinants such as race/ethnicity, education and sexual orientation (Tables 2 and 3), highlighting the fundamental significance of marital status in shaping the suicide risk among transgender people.

In most cases, we find that never married and previously married transmen and transwomen had higher risk of both suicide ideation and attempt than their married counterparts. Some of these differences in suicidality were partially explained by sociodemographic characteristics, and to a lesser extent by transition stage. Yet, most of the identified marital status differences in transgender suicidality remained significant after all covariates were accounted for. It is likely that having a legally recognized marriage is especially important for transgender people because they are more likely to lack economic, social, and psychological resources relative to the general population [21]. Presumably, marriage (primarily cisgender different-sex marriage and recently extended to same-sex marriage [44]) increases access to such protective resources (e.g., pooled income, social support, social integration) as suggested by a long-standing sociological literature [15, 16, 19]. In this sense, the marriage-equality movement and resulting policies to increase transgender people's access to legal marriage should be effective in reducing transgender people's suicide risk [14]. Another possibility is that transgender people who marry are more psychosocially and economically advantaged than transgender people who never marry or than those who experience separation, divorce, or widowhood. This advantage could protect against suicide risk even before entering into a union, suggesting a selection process [33]. Indeed, the processes of marriage selection may be more pronounced among the transgender population than the cisgender population given the variety of ways transgender individuals experience marital transitions (e.g., selection into marriage, marital

dissolution because of transgender related reasons) [13]. Because of the stigma and transphobia experienced within the marriage market, transgender people face major disadvantages in finding long-term partners and maintaining relationship stability. Therefore, transgender people who get and remain married may be highly selective based on their preexisting economic and sociopsychological advantages, which may explain the lower suicide risk among married transmen and transwomen relative to their never married or previously married counterparts.

Surprisingly, we found that never married transwomen, regardless of their partnership status, had lower risk of suicide ideation than their married transwomen counterparts once sociodemographic characteristics (e.g., age, race, education, income) were accounted for. This finding is unexpected. Indeed, without controlling for any covariates, never married transwomen, both partnered or unpartnered, had higher risk of both suicide ideation and attempt, but these differences were no longer present once sociodemographic characteristics were accounted for. Our descriptive statistics suggested that never married transwomen are less likely to identify as gay/lesbian (25% of partnered and 24% of unpartnered) than married transwomen (34%), and sexual minorities suffer higher risk of suicide [45, 46]—this may be a contributing factor for lower suicide risk of never married transwomen relative to married transwomen. Given cultural shifts around both gender/sexuality and marriage in recent decades [47, 48], it is also likely that unmarried transwomen, who in our sample are younger, feel less pressure to conform to normative expectations of family formation and gender/sexual identity and thus may experience less internalized gender-related stigma than married (and older) transwomen in our sample, stigma that can contribute to suicide risk [49]. Another possibility is that marriage, especially for transwomen who marry before coming out as transgender or transitioning, is particularly stressful and contributes to transgender-related sigma and suicidality [50]. This may be more relevant to transwomen's well-being than to transmen's because previous cisgender literature suggests that marital stress has a stronger negative impact on women's health and well-being than men's [51].

It is noteworthy that the identified marital status differences in suicidality among transgender people are more prevalent for past year suicide ideation than past year suicide attempt. It is important to note that our measure of attempt includes only respondents who reported ideation in the past year, a measure of attempt recommended by recent theoretical work on suicide [52, 53]. Specifically, the ideation-to-action framework suggests ideation and attempt have similar but also distinct correlates and pathways [52, 53]. According to the ideation-to-attempt framework, suicide ideation is due to a combination of psychological pain and hopelessness, and suicidal thoughts persist if social connections and support do not buffer feelings of pain and hopelessness. Ideation often leads to attempt when an individual with suicidal thoughts gains the capacity to go through with an attempt. Our results confirm previous theoretical and empirical work suggesting different correlates of suicide ideation and attempt, and our results extend examination of these differences to the transgender population. Among transmen, we find that marital status is a stronger positive correlate of ideation than of attempt among ideators—individuals who share dispositional characteristics correlated with suicide attempt. Marriage might be less strongly correlated with factors associated with attempt—factors such as proximal stressful events or the practical or dispositional capacity to go through with an attempt [52]—than with factors associated with ideation. While previous research among the general population suggests marital status has a similar association with ideation and attempt [54], this literature is limited and there is no work we are aware of that examines these unique associations among transgender individuals. Among transwomen, differences in results for ideation and attempt among ideators is more complicated: those who are married, relative to those who are never married, are *more* likely to report suicide ideation once socio-demographic factors are accounted for, yet there is no statistically significant difference between

married and never married individuals in suicide attempt among ideators once socio-demographic factors are accounted for. These results again highlight the relevance of the ideation-to-action framework when examining suicidality among the transgender population and the complex dynamics of marital status, coming out and transitioning, and suicidality, including how these dynamics vary between transmen and transwomen.

Our study is limited in several ways. First, the USTS did not use a nationally representative, population-based random sample. The recruitment process was based on convenience-sampling techniques (i.e., non-probability sampling methods). However, the USTS includes valuable information on suicidality and marital status among transgender people across the United States and is, so far, the most comprehensive large-scale dataset that addresses our research questions. Second, although we worked from a marital advantage and minority stress perspective to build our research hypotheses on how marital status shapes transgender people's risk of suicide, we could not determine causality or selection processes because of the cross-sectional nature of the data. Third, our measures of suicide ideation and attempt are limited by dichotomous responses. We are therefore unable to measure, for example, intensity of suicide ideation, which is positively associated with suicide attempt [55], self-reported likelihood of attempt, or how much participants think about suicide, limiting our ability to more fully understand the association between marital status and suicidality. The suicide measures available in the USTS data, however, are commonly used in empirical research on suicide, allowing for comparisons to studies using large national samples and to determine baseline levels of suicidality among the transgender population. The USTS data also allow us to measure past year suicide ideation among all transgender individuals as well as past year attempt among those with past year ideation, which better aligns with current understandings of suicidality [52]. Fourth, due to data limitation, we could not analyze all key predictors for suicide ideation and attempt such as cultural factors, personality, underlying stressful situations, comorbid psychiatric conditions and stress associated with one's current marriage or relationship. Future studies should explore the roles of these unobserved factors in shaping the risk of suicide among transgender people using other datasets to further understand whether these factors may explain the identified marital status differences in suicidality among transgender people. Finally, the USTS is lacking important information such as marital history, marital quality, marital duration, sexual orientation of the partner, and potential psychosocial mechanisms. All such information is important for understanding the life context of transgender people and their risk of suicide. Large-scale longitudinal data are needed that include more information on transgender people, preferably dyadic data that follows both transgender individuals and their partners.

## Conclusions

The suicide rate for transgender is among the highest of any group in the United States [4]. We analyzed one of the first currently available large-scale datasets of transgender people to provide an assessment of marital status differences in suicidality. Findings highlight marital status as a risk/protective factor for suicide among transmen and transwomen, with being unmarried (both never married and previously married) associated with higher risk of suicide (relative to being married) for transgender people. High levels of societal transphobia present a continued challenge for public policies and programs promoting marriage equality and equal treatment among the transgender population [56]. Our findings of marital status variation in suicide risk among transmen and transwomen draw attention to the heterogeneity of this population, highlighting marital status as a key social factor in stratifying the life experiences of transgender people. Public policies and programs should be designed and implemented at the

interpersonal and institutional levels in order to reduce suicide risk and other major disadvantages among transmen and transwomen, especially those who are unmarried.

## Author Contributions

**Conceptualization:** Hui Liu.

**Formal analysis:** Lindsey Wilkinson.

**Funding acquisition:** Hui Liu.

**Methodology:** Hui Liu.

**Writing – original draft:** Hui Liu.

**Writing – review & editing:** Lindsey Wilkinson.

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
