## [Decision Letter · Decision Letter 0]

10 May 2021

PONE-D-21-04050

Marital Status Differences in Suicidality among Transgender People

PLOS ONE

Dear Dr. Liu,

Thank you for submitting your manuscript to PLOS ONE. After careful consideration, we feel that it has merit but does not fully meet PLOS ONE’s publication criteria as it currently stands. Therefore, we invite you to submit a revised version of the manuscript that addresses the points raised during the review process.

We look forward to receiving your revised manuscript.

Kind regards,

Vincenzo De Luca

Academic Editor

PLOS ONE

Journal Requirements:

3.Thank you for stating the following financial disclosure:

Reviewers' comments:

Reviewer's Responses to Questions

**Comments to the Author**

1. Is the manuscript technically sound, and do the data support the conclusions?

Reviewer #1: Yes

Reviewer #2: Yes

Reviewer #3: Yes

Reviewer #4: Partly

2. Has the statistical analysis been performed appropriately and rigorously? 

Reviewer #1: Yes

Reviewer #2: Yes

Reviewer #3: Yes

Reviewer #4: I Don't Know

3. Have the authors made all data underlying the findings in their manuscript fully available?

Reviewer #1: Yes

Reviewer #2: Yes

Reviewer #3: Yes

Reviewer #4: Yes

4. Is the manuscript presented in an intelligible fashion and written in standard English?

Reviewer #1: Yes

Reviewer #2: Yes

Reviewer #3: Yes

Reviewer #4: Yes

5. Review Comments to the Author

Reviewer #1: This intriguing study sheds light on marriage and partnership as moderators of suicidal ideation and attempt among a large sample of transgender people. The manuscript is well-organized, easy to follow, and an interesting read. My two primary comments are in relationship to the discussion, which while well-written, could be expanded upon. First, I would like to see some discussion on suicidal ideation versus suicide attempt, particularly among the transwomen group. Second, it would be useful to discuss both effect sizes (small, moderate, large) and significance in the discussion section.

Minor comments:

First paragraph of introduction: Please provide statistics for suicidal ideation and attempt for cisgender people in the US, as a basis for comparison with transgender people.

In the description of covariates (p. 8) there are two places where (reference) is given. It seems as though the authors meant to insert references in these locations of the text.

Reviewer #2: I am very much enjoying reading the author's manuscript. This study assessed how marital status modifies the risk of suicide among transgender people. Below are my comments.

1. What is the meaning of “MICE”? Please provide the full name.

2. Authors divided marital status into five groups. However, I believe that readers will would like to know why authors do this classification.

3.Please provide the 95%CI for the results from logistic regression.

4. Pleas provide the specific P values.

5. How to define a transgender person? Self-report?

6. Please specifically describe the standard survey weight.

Reviewer #3: I appreciate the authors for taking up this important subject of research among transgender population. The introduction and the background of this topic is well justified and appropriate statistical analysis has been done. Discussion is also well written with good review of relevant literature. I have few queries and suggestions:

1)Since the data is from 2015 USTS ,how did the authors deal with the issue of consent from study participants to use their information in the current study?

2)participant's suicidal thought and attempt could be influenced by their personality, underlying stressful situations and comorbid psychiatric conditions

3)No information regarding whether their marital status(married or unmarried ) was perceived to be stressful or not.

4)Any information regarding the sexual orientation of the partners with whom the study population had been married or previously married. (it would be important to note because this factor may also influence the other protective factors in a married relationship status)

5) It would be also good to note the duration of married status and its relationship to suicidal ideation or attempt

Reviewer #4: - Suicidality in my opinion is very softly assessed, mainly on the basis of questions with binary answers. This is a weakness the authors must acknowledge and explain in the manuscript.

- It would be prudent to include definitions of transmen and transwomen to make it easy for novice readers in this field.

- Though a few covariates have been rightly included, certain others such as cultural factors, co-morbid mental illness etc have not been included and this deserves mention in the limitations section.

Overall i feel a more comprehensive assessment of sucidality in a more limited representative population is better suited to the research question at hand.

6. PLOS authors have the option to publish the peer review history of their article (what does this mean?). If published, this will include your full peer review and any attached files.

Reviewer #1: No

Reviewer #2: No

Reviewer #3: No

Reviewer #4: **Yes: **Kartik Singhai

---

## [Author Response · Author response to Decision Letter 0]

18 Jun 2021

RESPONSE MEMO

We greatly appreciate the helpful comments and suggestions from the reviewers. In the following memo, we make a point-by-point response to detail how we incorporated comments made by each reviewer into the revised manuscript. The text in quotations are the direct quotes from the reviewers’ comments, which are followed by our response. 

Response to Reviewer 1

1. “First, I would like to see some discussion on suicidal ideation versus suicide attempt, particularly among the transwomen group.” 

We now discuss differences in results between suicidal ideation and attempt (pp. 15-16), emphasizing that our measure of attempt includes only ideators. We discuss the implications of this measurement of ideation and attempt, integrating the ideation-to-action theoretical framework, highlighting our contribution to this framework and indicating the need for future work to explore, separately, correlates of ideation and attempt among ideators. Differences in findings for ideation and attempt among transmen and transwomen further highlight the need for more understanding of the dynamic interplay of marriage timing and the timing of transition, a dynamic that likely affects the association between marital status and suicidality and likely varies among transmen and transwomen. 

2. “Second, it would be useful to discuss both effect sizes (small, moderate, large) and significance in the discussion section.”

We have added some discussion about effect sizes of marital status in the discussion section (p. 13). Specifically, we compared the sizes of marital status effects to other fundamental factors of social determinants such as race/ethnicity, education and sexual orientation and discussed the implications (p. 13). 

3. “First paragraph of introduction: Please provide statistics for suicidal ideation and attempt for cisgender people in the US, as a basis for comparison with transgender people.”

We have provided the relevant statistics for the general U.S. population (p. 3).

4. “In the description of covariates (p. 8) there are two places where (reference) is given. It seems as though the authors meant to insert references in these locations of the text.”

We apologize for the confusion. The “reference” on page 8 was to indicate the reference group in the analysis. We have now clarified it (p. 8).

Response to Reviewer 2

1. “What is the meaning of “MICE”? Please provide the full name.”

We now spell out the full name of MICE, Multiple Imputation using Chained Equations (p. 7).

2. “Authors divided marital status into five groups. However, I believe that readers will would like to know why authors do this classification.”

We now provide a justification for our approach to categorize marital status groups (p. 8).

3. “Please provide the 95%CI for the results from logistic regression.” 

We have now included 95% CIs in logistic regression results.

4. “Pleas provide the specific P values.”

We have now added specific p-values to logistic regression results.

5. “How to define a transgender person? Self-report?”

We now clarify that our transgender respondents are self-identified as transgender (p. 7). We provide a clear definition of transgender (i.e., assigned birth sex is different from current primary gender identity and/or respondent identifies primarily as transgender or transsexual) for the purpose of clarifying our study sample, although we understand defining the transgender population is challenging given that there is no universally agreed definition of the term, transgender.

6. “Please specifically describe the standard survey weight.”

The standard survey weight incorporates a weight constructed using information from the American Community Survey to help adjust for hypothesized over-representation of whites and the 18-year-old group in the USTS sample. We now include this information on p. 10.

Response to Reviewer 3

1) “Since the data is from 2015 USTS ,how did the authors deal with the issue of consent from study participants to use their information in the current study?”

This study is a secondary data analysis of public dataset and no individual can be identified in the data. This study was approved by the Michigan State University institutional review board.

2) “participant's suicidal thought and attempt could be influenced by their personality, underlying stressful situations and comorbid psychiatric conditions.”

Unfortunately, we don’t have these variables in our data. We acknowledge this limitation and discuss this point and call future research to explore these variables using other datasets (p. 15).

3) “No information regarding whether their marital status(married or unmarried ) was perceived to be stressful or not.”

Our analysis is limited by our data, which does not provide information about whether their marital status was perceived to be stressful or not. We acknowledge this limitation and discuss this point and call future research to explore this possibility using other datasets (p. 17).

4) “Any information regarding the sexual orientation of the partners with whom the study population had been married or previously married. (it would be important to note because this factor may also influence the other protective factors in a married relationship status).”

Unfortunately, we don’t have information regarding the sexual orientation of the partner. We acknowledge this limitation and call for more data collection in this field to include the partner’s information (p. 17).

5) “It would be also good to note the duration of married status and its relationship to suicidal ideation or attempt.”

Unfortunately, we don’t have information about duration of marital status in our data. We acknowledge this limitation and discuss this point and call more data collections (pp. 17).

Response to Reviewer 4

1. “Suicidality in my opinion is very softly assessed, mainly on the basis of questions with binary answers. This is a weakness the authors must acknowledge and explain in the manuscript.”

While binary measures of suicide are limited in their ability to assess, for example, intensity of suicide ideation, the measures available in the USTS data are commonly used in empirical research on suicide using large national samples, allowing for comparisons across studies utilizing similar measures. While the binary measures provided in the USTS present weaknesses, we have taken advantage of the ability to measure past year suicide ideation among all transgender individuals as well as past year attempt among those with past year ideation, which better aligns with current understandings of suicidality. We add discussion and acknowledgement of the limitations of our measures for suicidality and call for collection of additional measures of suicidality among large-scale datasets surveying the transgender population (pp.15-17). Nevertheless, the 2015 U.S. Transgender Survey (USTS), conducted by the National Center for Transgender Equality and the National Gay and Lesbian Task Force, is so far the best dataset available to address our research questions. There are no nationally representative population-based data representing transgender Americans. 

2. “It would be prudent to include definitions of transmen and transwomen to make it easy for novice readers in this field.”

As suggested, we include definitions of transmen and transwomen on page 7.

3. “Though a few covariates have been rightly included, certain others such as cultural factors, co-morbid mental illness etc have not been included and this deserves mention in the limitations section.”

Unfortunately, we don’t have these variables in our data. We acknowledge this limitation and discuss this point and call future research to explore these variables using other datasets (p. 17).

4. “Overall i feel a more comprehensive assessment of suicidality in a more limited representative population is better suited to the research question at hand.”

We agree with the reviewer on the limitations of the measures and representativeness of the data. However, the 2015 U.S. Transgender Survey (USTS), conducted by the National Center for Transgender Equality and the National Gay and Lesbian Task Force, is so far the best dataset available to address our research questions. There are no nationally representative population-based data representing transgender Americans. Although the measures of suicidality are limited, they are generally consistent with other national suicide studies 

(e.g., The National Institute of Mental Health. Suicide. [cited 2020 September]. Available from: https://www.nimh.nih.gov/health/statistics/suicide.shtml). We also acknowledge such limitations in the paper (pp. 16-17). Given the paucity of research in this area using large datasets, we believe the strengthens and potential scientific and public significance of this study outweigh its limitations. We hope our effort in this line of research would inspire more future research and more data collection in this emerging field.

---

## [Decision Letter · Decision Letter 1]

19 Jul 2021

Marital Status Differences in Suicidality among Transgender People

PONE-D-21-04050R1

Dear Dr. Liu,

We’re pleased to inform you that your manuscript has been judged scientifically suitable for publication and will be formally accepted for publication once it meets all outstanding technical requirements.

Kind regards,

Vincenzo De Luca

Academic Editor

PLOS ONE

Additional Editor Comments (optional):

Reviewers' comments:

Reviewer's Responses to Questions

**Comments to the Author**

1. If the authors have adequately addressed your comments raised in a previous round of review and you feel that this manuscript is now acceptable for publication, you may indicate that here to bypass the “Comments to the Author” section, enter your conflict of interest statement in the “Confidential to Editor” section, and submit your "Accept" recommendation.

Reviewer #1: All comments have been addressed

Reviewer #2: All comments have been addressed

Reviewer #3: All comments have been addressed

Reviewer #4: All comments have been addressed

2. Is the manuscript technically sound, and do the data support the conclusions?

Reviewer #1: Yes

Reviewer #2: Yes

Reviewer #3: (No Response)

Reviewer #4: Yes

3. Has the statistical analysis been performed appropriately and rigorously? 

Reviewer #1: Yes

Reviewer #2: Yes

Reviewer #3: (No Response)

Reviewer #4: Yes

4. Have the authors made all data underlying the findings in their manuscript fully available?

Reviewer #1: Yes

Reviewer #2: Yes

Reviewer #3: (No Response)

Reviewer #4: Yes

5. Is the manuscript presented in an intelligible fashion and written in standard English?

Reviewer #1: Yes

Reviewer #2: Yes

Reviewer #3: (No Response)

Reviewer #4: Yes

6. Review Comments to the Author

Reviewer #1: (No Response)

Reviewer #2: Thank you for your revision. Only one issue need to be addressed.

Please re-check all the tables and use "P<0.001" to replace "P=0.000"

Reviewer #3: (No Response)

Reviewer #4: (No Response)

7. PLOS authors have the option to publish the peer review history of their article (what does this mean?). If published, this will include your full peer review and any attached files.

Reviewer #1: No

Reviewer #2: No

Reviewer #3: **Yes: **Manoj Prithviraj

Reviewer #4: **Yes: **Kartik Singhai

---

## [Editor Report · Acceptance letter]

25 Aug 2021

PONE-D-21-04050R1 

Marital Status Differences in Suicidality among Transgender People 

Dear Dr. Liu:

I'm pleased to inform you that your manuscript has been deemed suitable for publication in PLOS ONE. Congratulations! Your manuscript is now with our production department. 

Kind regards, 

on behalf of

Dr. Vincenzo De Luca 

Academic Editor

PLOS ONE